# Ambiental Factors in Parkinson’s Disease Progression: A Systematic Review

**DOI:** 10.3390/medicina59020294

**Published:** 2023-02-05

**Authors:** Anastasia Bougea, Nikolas Papagiannakis, Athina-Maria Simitsi, Elpida Panagiotounakou, Chrysa Chrysovitsanou, Efthalia Angelopoulou, Christos Koros, Leonidas Stefanis

**Affiliations:** 1st Department of Neurology, Medical School, National and Kapodistrian University of Athens, Eginition Hospital, 11528 Athens, Greece

**Keywords:** Parkinson’s disease (PD), air pollutants, organic matter (OM), UV light, levodopa equivalent dose (LED), *Unified Parkinson’s Disease Rating Scale* (UPDRS), non-motor symptoms, hallucination

## Abstract

*Background and Objectives*: So far, there is little evidence of the ambient effect on motor and non-motor symptoms of Parkinson’s Disease (PD). This systematic review aimed to determine the association between ambiental factors and the progression of PD. *Materials and Methods:* A systematic literature search of PubMed, Cochrane, Embase, and Web of Science was conducted up to 21 December 2021 according the Preferred Reporting Items for Systematic Reviews and Meta-Analyses (PRISMA) guidelines. *Results*: Eight articles were used in the analyses. Long-term exposure to fine particles (particulate matter ≤ 2.5 μm; PM_2.5_) was positively associated with disease aggravation in two studies. Short-term PM_2.5_ exposure was positively associated with disease aggravation in three studies. Significant associations were found between PD aggravation and NO_2_, SO_2_, CO, nitrate and organic matter (OM) concentrations in two studies. Associations were more pronounced, without reaching statistical significance however, in women, patients over 65 years old and cold temperatures. A 1% increase in temperature was associated with a significant 0.18% increase in Levodopa Equivalent Dose (LED). Ultraviolet light and humidity were not significantly associated with an increase in LED. There was no difference in hallucination severity with changing seasons. There was no evidence for seasonal fluctuation in Unified Parkinson’s Disease Rating Scale (UPDRS) scores. *Conclusions*: There is a link between air pollutants and temperature for PD progression, but this has yet to be proven. More longitudinal studies are warranted to confirm these findings.

## 1. Introduction

Parkinson’s disease (PD) is a progressive neurodegenerative disease with motor and non-motor symptoms characterized by complex interactions between genetic and environmental factors [1]. These factors influence the onset and varying evolution rate of PD, however the exact mechanisms remain unclear [1,2]. Given the ineffective symptomatic treatments of PD, the research priority is focused on prevention and the recognition of modifiable environmental risk factors including pesticides, heavy metals, viruses, and air pollution [3]. Importantly, air pollution—unlike other putative modifiable risk factors—could be reduced at the population level through cooperative action, environmental legislation, and green technological invention [4].

Particulate matter (PM) consists of solid or liquid particles in the air of varying sizes in the following categories: PM_10_ (particles with diameter < 10 μm); PM_2.5_ or fine particles (<2.5 μm); coarse particles, which complement fine particles that are defined by diameter between 2.5 and 10 µm; and ultrafine particles or PM_0.5_ (<0.5 μm) [5]. Previous studies have suggested an association between particulate air pollution and PD, highlighting the role of potential biological pathways such as oxidative stress and inflammation in neuronal loss and dopaminergic degeneration [6,7,8,9]. The inhalation of neurotoxic PM increased the α-synuclein (α-Syn) aggregation among mice models, providing mechanistic insights into how PM increases the risk for PD [10,11].

Although no definitive evidence has been established, there are several potential causal pathways linking temperature, ultraviolet light, and humidity to PD [12,13,14]. In rat models, exposure to low temperatures slowed the firing and cell membrane hyperpolarization, increased cell input resistance, generated an outward current under voltage clamp and decreased Ca^2+^, while high temperatures induced the opposite effects [15]. Cold exacerbates motor symptoms of PD patients due either to shivering thermogenesis or to a decrease of muscular temperature [14]. Seasonal temperature and humidity are associated with PD medications [12]. A nationwide ecologic study demonstrated an age-dependent association between ultraviolet light and PD incidence [16].

Although, there are currently several systematic reviews and more than 500 studies on environmental factors (i.e., metals, pesticides, paraquat, air pollutants, and rural farming) and the associated risk for PD onset, few studies are focused on the evidence of the environmental exposures that influence disease progression [17,18,19,20,21,22,23].The present systematic review aims to investigate environmental factors, such as chemicals or toxic emissions, temperature, ultraviolet light and humidity, that influence PD progression/severity. Moreover, the present review focused only on PD and did not include other disorders such as multiple system atrophy and dementia with Lewy bodies that follow a similar pathophysiology with PD, because it was performed in the context of the ALAMEDA project that proposes a novel combination of minimally intrusive wearable devices and smartphone applications required for the desired analysis of motor and non-motor symptoms of PD patients.

## 2. Materials and Methods

### 2.1. Search Strategy

This review has been registered in the International Prospective Register οf Systematic Reviews (PROSPERO) ID: CRD42022307417.The preferred Reporting Items for Systematic Reviews and Meta-analysis (PRISMA) guidelines were followed [24].

We performed a systematic literature search of the following databases: MEDLINE, EMBASE, CINAHL, PsyclinINFO, Scopus, and Web of Science. A search was carried out to identify studies published up to 21 December 2021 (no start date was given in order to cover as many publications as possible) that evaluated the ambiental factors in PD. The search terms combined Medical Subject Headings (MeSH) terms and text words including: “PM_2.5_”“PM_10_” OR “PM_(10)_” OR “fine particulate matter” OR “fine particles” OR “nitrogen oxides” OR “nitrogen dioxide” OR “NO ” OR“NO_2_” OR “Sulfate”, OR “carbon monoxide” OR “CO”, “air pollution” OR “air pollutants” OR “humidity”, “temperature”, OR“UV light”, and “Parkinson” OR “Parkinson’s disease”, OR “PD” OR “Parkinsonism” OR “Parkinsonian” OR “Parkinson’s disease aggravation” OR “Parkinson’s disease progression”. The search strategy was detailed in the Appendix A.

### 2.2. Inclusion Criteria

Inclusion criteria were publication language should be English, and the included subjects should have a clinical PD diagnosis based on standardized criteria (UK Brain Bank) [25] at baseline. Additionally, only longitudinal prospective cohorts were considered, which evaluated at least one ambient factor in relation to the progression or severity of PD.

### 2.3. Exclusion Criteria

We excluded all studies that did not consider the relationship between ambiental factors and PD progression/severity or those that consider similar disorders with PD such as multiple system atrophy and dementia with Lewy bodies, conference abstracts or publications without data. Animal model studies were also excluded.

### 2.4. Data Extraction and Data Measurement

The following variables from the included articles were collected in a standardized extraction form: first author name, year of publication, study design, sample size, age- and sex-adjusted odds ratio (OR) or hazard ratio (HR) and 95% confidence interval, and number of PD cases. If risk estimates were unavailable, the corresponding author of included articles was contacted to provide them. Any disagreements were resolved through discussion and consensus or consultation with the authors. Data collection was carried out independently by two reviewers and disagreements were discussed with a third reviewer.

### 2.5. Assessment of the Methodological Quality of the Included Studies

Quality assessment was performed for each study using the modified Newcastle–Ottawa scale for cohort studies [26]. A score of 0–9 was allocated to each study. Studies with 9 were considered high quality, 8–7 points indicated moderate quality, and less than 6 showed low quality.

## 3. Results

### 3.1. Description of Selected Studies

The search strategy initially provided 13, 216 eligible studies. After removing duplicates, 3383 studies were suitable for initial screening and, by reading the title and abstract, 690 publications were considered candidates for review in accordance with the previously established inclusion criteria. After full text review, eight articles were selected for review; the flowchart of the study selection is shown in Figure 1.

Five out of the seven studies had a moderate quality (NOS score of 8), as seen in Table 1.

We only conducted qualitative syntheses, because of the small number of studies and great heterogeneities in the study populations, exposure assessment methods, and covariate adjustments.

### 3.2. Air Pollution Exposure

This part of the review focuses on the short- or long-term effects of the individual pollutants on PD, including particulate matter (PM_10_, PM_2.5_ and PM_2.5–10_), NO_x_, NO_2_, CO, organic matter (OM), black carbon (BC), sulfate, sea salt (SS), and soil particles. Table 2 outlines the relationships between these pollutants and PD identified in the studies included in this review.

For each 10 μg/m^3^ increase in the two-day average of PM_2.5_, Zanobetti et al. [27] found a significant association between short-term exposure to PM_2.5_ and all-cause mortality in subjects with previous hospital admissions for diabetes (0.76%; 0.39, 1.12), PD (1.15%; 0.09, 2.23), dementia (0.94%; 0.01, 1.89), and Alzheimer’s disease (1.04%; 0.36, 1.72). However, these associations were not significantly different from the mortality risk among Medicare enrollees that were never hospitalized for any of these diseases. Kioumourtzoglou et al. [28] noted an 8% increase in hazard of hospital admission of PD patients for every one μg/m^3^ increase in annual PM_2.5_ citywide concentrations (95% CI: 1.04, 1.12, *p* < 0.05). Lee et al. [29] showed that PD aggravation was significantly associated with an increase in the 8-day moving average ofPM_2.5_ (odds ratio: 1.61 [1.14–2.29] per 10  μg/m^3^), NO_2_ (2.35 [1.39–3.97] per 10 ppb), SO_2_ (1.54 [1.11–2.14] per 1 ppb), and CO (1.46 [1.05–2.04] per 0.1 ppm). Although the associations were more pronounced in women, patients over 65 years old and cold temperatures, they were not statistically significant. In the Shi study [30] there was a 13% increase in hazard of hospital admission for PD for every 5 μg/m^3^ increase in annual PM_2.5_ concentrations(95% CI 1.12–1.14). In the Nunez prospective study [31] elevated annual OM and nitrate levels were independently associated with higher first PD hospitalization rates. A nonlinear negative association was detected between BC at concentrations above the 96th percentile and PD aggravation. No correlation was detected with sulfate, SS and soil exposure.

### 3.3. Temperature, Humidity, Solar Exposure and Seasonal Changes

This portion of the review examined the impact of the ambiental factors on PD, including temperature, humidity, solar exposure, and seasonal changes. Table 3 describes the relationships between these ambiental factors and PD in the studies included in this review.

Postuma et al. [32] explored whether severity of PD, as measured by the Unified Parkinson’s Disease Rating Scale (UPDRS)I-IV, changed with the season (at least two evaluations of at least one UPDRS score). There was no significant seasonality in any UPDRS subscale. The UPDRS II and III scores were highly constant throughout the year. In the Goetz study, [33] there was no association in hallucination severity with seasonal swift; the level of wintertime aggravation was no greater than summertime exacerbation (P0.42). Although hallucinations often evolve in dark settings and evening hours, the darkness of wintertime does not aggravate hallucinations in PD patients on a stable regimen. Rowell et al. [12] tested what effect the temperature, humidity, and solar exposure may have on PD medications, measured by the levodopa equivalent dose (LED). The prescribed LED was 7.4% greater in January and 8% lower in July. Temperature was associated with the prescription of PD medications.

## 4. Discussion

The present systematic review suggests a potential relationship between environmental factors and progression of PD. A number of air pollutants—PM_2.5_, NO_2,_ SO_2,_ CO, nitrate, and OM—were identified to be associated with progression of PD. Humidity but not UV light was significant associated with an increase in LED. There was no difference in hallucination severity with changing season (the darkness of wintertime does not exacerbate hallucinations in PD subjects on stable medications). There was no evidence for seasonal fluctuation in the UPDRS scores.

Animal and autopsy studies have explored several pathways by which air pollutants lead in neurotoxicity of PD [34,35]. Air pollution could be associated with PD onset through brain inflammation and oxidative stress [7,36,37], or PD progression [38,39]. Airborne ultrafine particulates are transferred into the blood–brain barrier (BBB) via the olfactory cortex [34,40,41]. Intrastriatal injection or intranasal administration of PM_2.5_ exacerbates α-Syn pathology and dopaminergic neuronal degeneration in α-Syn A53T transgenic mice [11]. Although difficult to interpret, animal-to-human translation could be the key to clarifying neurodegenerative mechanisms. Nevertheless, the lack of an association does not automatically entail the absence of a causal relationship. There are several methodological differences (study design, exposure method or duration, and variable accuracy of regional air pollution monitoring), that are attributed to inconsistent results, emphasizing the necessity for more comprehensive studies. Epidemiologic studies are commonly used to supplement controlled studies of acute effects of short-term exposure, and susceptibility to confounding limits the certainty of the results.

Another interesting point of this review was the absence of seasonal variation in the severity of PD in terms of motor deterioration or hallucinations [32,33]. The evident reason is that the overall state of a parkinsonian patient is untouched by seasonal factors. A second is that patients may adapt to seasonal changes by dosage alterations at one’s discretion. Finally, two factors antagonize each other; for instance, longer sleep might benefit function in the winter, but poor exercise cancel out this benefit. However, these studies only included small samples for a short follow-up period. In contrast, Rowell et al. [12] followed up 70,000 people with PD from Australia for 23 years of monthly pharmaceutical use. The seasonal variation in the LED concurred with clinical findings described by Kaasinen et al. [42] Elevated dopamine synthesis in PD patients during winter results in low prescribed LED, and, conversely, a decrease in dopamine during summer leads to augmented LED. Nevertheless, this is an ecological study that should be handled as exploratory rather than explanatory. Therefore, we cannot absolutely exclude the possibility that other seasonal environmental and behavioral factors under the influence of temperature could be affecting the LED. Undoubtedly, well-designed studies, (such as longitudinal), are needed to confirm the findings.

The present review has interesting clinical and research implications. Most studies cover both urban and rural areas, as well as a diverse population (e.g., SPARCS includes information on hospitalizations of all ages and independently of health insurance in Nunez et al., 2021)use of flexible models that allow the testing of both linear and nonlinear associations to characterize the exposure–response relationships better. There is a necessity to consider the residential or indoor pollutant sources, as well as occupational exposures such as workplaces [43,44], in which case personal monitoring or biomonitoring may be more suitable to gauge individual exposure accurately amidst increasingly mobile populations. Multipollutant models are recommended to understand the interaction and synergistic effects between different air pollutants better [45]. The effects of lesser-studied pollutants such as SO_2_ and CO also warrant further examination.

Although the quality assessment of studies included was adequate, there were several limitations. First the cohort size was variable and study design was poor, such as ecological, (only three longitudinal studies). Second, the heterogeneity of data (windows of exposure, lack of adjustment for confounders, absence of sensitive measurement tools, exposure measurement error, and follow-up duration) was a major confounding factor. UPDRS was not sensitive to detect seasonal variation of non-motor symptoms such as hallucinations, and depression [32,33]. These two studies did not adjust for potential confounding factors that may determine such variations, such as exercise, mood, sleep, and diet. In the same line, another study suggested that an unobserved seasonal variable would need to be highly correlated with temperature to confound the association between temperature and LED [12]. However, the latter study followed up 70,000 patients with PD for 23 years [12], while the other two studies followed up comparatively small samples for a short time (2 or 3 years) [33]. Two studies cannot exclude the possibility of potential residual confounding bias [30,31], however in another study it is not likely to have occurred [28]. Third, the majority of the studies defined PD aggravation as emergency admissions for primarily diagnosed PD without direct measurement of aggravation of PD symptoms, due to limited data availability. In these studies, misclassification of exposure is more likely to result from including patients who do not experience disease aggravation, but are hospitalized due to unrelated health issues [27,28,29]. By contrast, one study used PM_2.5_ as a sensitive indicator of the disease aggravation [31]. Fourth, there is an unanswered question as to whether the types of air pollutants examined are the right ones to study with regard to PD. In particular, the relationship between particle size and their impact on PD is unclear. For example, do ultra-fine particulates present a greater risk than larger particles? Are the particles (PM_10_ and PM_2.5_) studied to date too large to be related to PD risk and should we be focusing future studies on smaller, including ultrafine, particles? The evidence of sex differences to air pollution and PD progression/severity remains unknown, and further investigation and reporting of sex-stratified results will be informative and may shed light into possible biological mechanisms.

This systematic review has its limitations. There is lack of studies on ambiental factors related to PD progression. As detailed above, the methodological diversity of the included studies requires attention in interpreting the synthesized results. We were not able to perform a meta-regression due to the high heterogeneity between the studies. The lack of data on several clinical outcomes could influence the results.

However, the present systematic review also presents positive points: a strict methodology was applied for the literature search to encompass the latest evidence; restriction of inclusion and exclusion criteria were defined for study selection and a validated tool was used to assess the quality of included studies.

## 5. Conclusions

The contribution of ambient factors to the progression in PD is still largely unexplored. Studies indicating that air pollutants, temperature, and humidity are associated with PD progression are of limited generalizability because of methodological flaws. Further research on the effect of ambient factors on severity would also help to determine whether these factors may represent effective targets for policies that result in further reductions of air pollutant concentrations. More studies should explore the link and toxicological mechanisms associated with gene–environment interaction and air pollution. Keeping in mind that the brain is the most heat-sensitive organ, identifying how temperature may affect the motor and non-motor symptoms of PD would improve our understanding of the pathophysiology of PD. Future studies could examine the interaction between the human gut microbiome and exposure to air pollution with regard to PD risk and possibly progression. Dose–response effects evaluated by high-quality studies are warranted, especially on seasonal effect ultraviolet light on PD progression or severity. Our review also underlines the importance of longitudinal studies that assess factors for which evidence is inconclusive. Another gap that needs to be filled is the limited body of evidence available on the relative strength of factors for PD severity: the simultaneous analysis of various potential risk factors would help to fill this gap. Now is the time to take advantage of the multipollutant statistical methods, such as machine learning algorithms, to predict the health effects of air pollution on PD progression. However, by employing a wide variety of multipollutant statistical methods across different epidemiologic study designs, we build up the essential scientific base in order to potentially develop more sustainable, multipollutant air quality regulations. Toward this direction, biomonitoring with plants would bean adequate alternative technique to acquire data about pollution exposure to residential or occupational ambient factors of patients with PD.

## Figures and Tables

**Figure 1 medicina-59-00294-f001:**
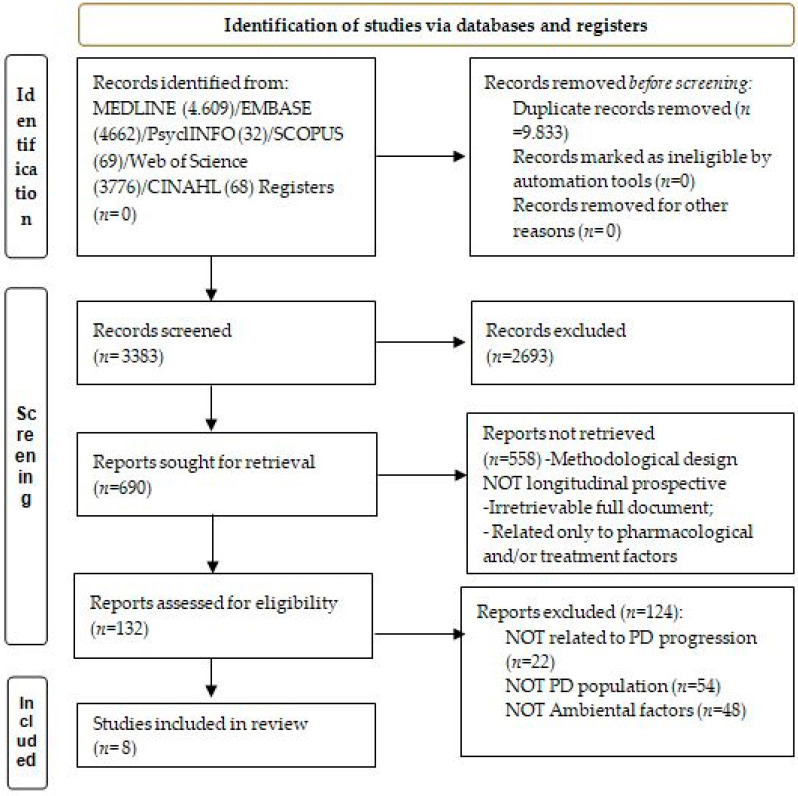
PRISMA 2020 Flowchart.

**Table 1 medicina-59-00294-t001:** Overviewof air pollutants and their relationship with PD progression.

Article	Representativeness	Selection of Non-Exposed Cohort	Ascertainment of Data Collection	PD Present at Start	Comparability (Adjustments)	Outcome Assessment	Duration of Follow-Up	Adequacy of Follow-Up	Total Score
Zanobetti, 2014; [27]	1	1	1	1	2	1	1	0	8
Kioumourtzoglou, 2016; [28]	1	1	1	1	2	1	1	0	8
Lee, 2017; [29]	0	1	1	1	1	1	1	0	6
Shi, 2020; [30]	1	1	1	1	2	1	1	0	8
Nunez, 2021; [31]	1	1	1	1	2	1	1	0	8
Postuma, 2005; [32]	1	1	1	1	2	1	1	0	8
Goetz, 2006; [33]	0	1	1	1	2	1	0	0	6

Rowell, 2017 was not included because this is an ecological study. Representativeness: 1 = truly representative of general population or somewhat representative; 0 = selected group of patients. Selection of non-exposed cohort: 1, drawn from the same community as the exposed cohort. Ascertainment of data collection: 1 = (thorough) healthcare record or check; 0 = no specific record. PD present at start: 1 = yes; 0 = parkinsonian traits present. Comparability (adjustments): 1 = study controls for age and sex; 2 = study controls for additional factors. Outcome assessment: 1 = independent blind assessment; record linkage; 0 = self-report; other or no description. Duration of follow-up: 1 = median duration of follow-up ≥ 3 years; 0 = median duration of follow-up ≤ 3 years. Adequacy of follow-up: 1 = complete follow-up, all subjects accounted for; subject lost ≤ 30% or description of lost-to-follow-up suggests no (potential) bias was introduced; 0 = follow-up rate < 70% and no description of cases lost to follow-up; no statement.

**Table 2 medicina-59-00294-t002:** Overview of air pollutants and their relationship with PD progression.

Authors, Year	Country	Study Type	Ambient Factor	N° PD Cases	PD Progression Measure	Follow-Up, Year	RR or HR	95% CI
Zanobetti, 2014; [27]	121 US communities (1999–2010)	case-crossover	short-term exposure to PM_2.5_ ^1^	NR	PDhospital admissions	11	3.23	1.08, 5.43
Kioumourtzoglou, 2016; [28]	50 northeastern U.S. cities (1999–2010)	cohort	long-term PM_2.5_ ^2^	NR	PDhospital admissions	11	1.08	1.04, 1.12
Lee, 2017; [29]	North Seoul	cohort	short-term exposure to PM_2.5_ ^1^, NO_2,_ SO_2_, CO	77	PDhospital admissions	11	PM_2.5_: 1.61 NO_2_: 2.35 SO_2_: 1.54 CO: 1.46	PM_2.5_: 1.14–2.29 NO_2_: 1.39–3.97 SO_2_: 1.11–2.14 CO: 1.05–2.04
Shi, 2020; [30]	USA(2000–2016)	Longitudinal-cohort	short-term exposure to PM_2.5_ ^3^	10^6^	PDhospital admissions	16	1.13	1.12–1.14
Nunez, 2021; [31]	62 NYS counties (2000–2014)	cohort	long-term PM_2.5_ ^4^, Nitrate, OM, black carbon, sulfate, soil particles	197, 545	PDhospital admissions	14	Nitrate: 1.06 OM: 1.06	PM_2.5_: Nitrate: 1.03–1.10 OM: 1.04–1.09

PM_2.5_ ^1^: 10 μg/m^3^ increasein the two-day average concentrations of PM_2.5_, PM_2.5_ ^2^: 1-μg/m^3^ increase in annual PM_2.5_ city-wide exposure, PM_2.5_ ^3^: 5 μg/m^3^ increase in annual PM_2.5_ concentrations, PM_2.5_ ^4^: increase from 8.1 to 104 mg/m^3^ increase, NR: no reported.

**Table 3 medicina-59-00294-t003:** Temperature, humidity, solar exposure, seasonal changes and PD progression.

Author, Year	Country	Study Type	Ambient Factor	N° PD Cases	PD Progression Measure	Follow-Up Time, Years	Main Findings
Rowell, 2017; [12]	Australia (eightstates *)	Ecological	Temperature, humidity and solar exposure	NR	Changes in the aggregate (LED)	23	The prescribed LED was 7.4% greater in January and 8% lower in July. Temperature but not UV lightand humiditywas associated with the prescription of PD medications.
Postuma, 2005; [32]	Canada (Toronto)	Longitudinal cohort	Fourseasons	546	UPDRS I-III off/on score in two evaluations	3	No significant seasonal variation in any UPDRS subscale
Goetz, 2006; [33]	USA (Chicago)	Longitudinal cohort	Two seasons	51 PD with hallucination	Thought Disorder (TD) score for assessment of hallucinations	1	The level of wintertime exacerbation was no greater than summertime exacerbation. No seasonal change in hallucination severity of PD patients.

* Sydney, Melbourne, Brisbane, Adelaide, Perth, Hobart, Canberra, Darwin, LED: levodopa equivalent dose.

## Data Availability

Not applicable.

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
