# Peer review of "Ambiental Factors in Parkinson’s Disease Progression: A Systematic Review"

_medicina, 2023, doi:10.3390/medicina59020294_

Round 1

Reviewer 1 Report

In this study, the authors report systematic review and meta-analysis of ambient effects on PD. Long-term exposure to fine particles (PM2.5) was positively associated with disease aggravation in 2 studies. Short-term PM2.5 exposure was positively associated with disease aggravation in 3 studies. Significant associations were found between PD aggravation and NO2, SO2, CO, nitrate and organic matter (OM) concentrations in 2 studies. Associations were more pronounced, without reaching however statistical significance, in women, patients over 65 years old and cold temperatures,. A 1% increase in temperature was associated with a significant 0.18% increase in Levodopa Equivalent Dose (LED). Ultraviolet light and humidity were not significantly associated with an increase in LED. There was no difference in hallucination severity with changing season. There was no evidence for seasonal fluctuation in Unified Parkinson's Disease Rating Scale (UPDRS) scores. In conclusion, specific air pollutants and temperature variations are significant factors for PD progression although more longitudinal studies are warranted to confirm these findings. This study is of interest for the readers of the journal, and is well written. I recommend that the current article is suitable for publication.

Author Response

Reviewer 1

In this study, the authors report systematic review and meta-analysis of ambient effects on PD. Long-term exposure to fine particles (PM2.5) was positively associated with disease aggravation in 2 studies. Short-term PM2.5 exposure was positively associated with disease aggravation in 3 studies. Significant associations were found between PD aggravation and NO2, SO2, CO, nitrate and organic matter (OM) concentrations in 2 studies. Associations were more pronounced, without reaching however statistical significance, in women, patients over 65 years old and cold temperatures,. A 1% increase in temperature was associated with a significant 0.18% increase in Levodopa Equivalent Dose (LED). Ultraviolet light and humidity were not significantly associated with an increase in LED. There was no difference in hallucination severity with changing season. There was no evidence for seasonal fluctuation in Unified Parkinson's Disease Rating Scale (UPDRS) scores. In conclusion, specific air pollutants and temperature variations are significant factors for PD progression although more longitudinal studies are warranted to confirm these findings. This study is of interest for the readers of the journal, and is well written. I recommend that the current article is suitable for publication.

OUR RESPONSE: Thank you for your valuable comment.

Reviewer 2 Report

Bougea et al present a systematic review of the role of environmental factors in Parkinson's Disease progression.  The authors identified 8 suitable articles and evaluated the following variables' impact on disease progression: exposure to particulate matter of varying sizes, exposure to inhaled compounds, temperature, UV light exposure, and humidity.  While this is certainly a very interesting topic, the very small size of the systematic review (further complicated by poor study quality of the papers included) limits the generalizability and validity of this review.  Please consider the following comments:

1. The Fig. 1 PRISMA flowchart indicates that 558 records were not retrieved.   Why?  It's ok to exclude reports in accordance to criteria, but those criteria must be justified beyond "not retrieved"... especially if the final study size is only 8 papers.

2. With such a small number of papers  specific to PD, and given that the hypothesized mechanism of action central to the investigation of environmental factors and PD involves the exacerbation of protein aggregation, it would be prudent to broaden the scope of the investigation to include other disorders that follow a similar pathophysiology to validate its replicability or perhaps to identify a differentiating factor for PD.  

Author Response

Comments and Suggestions for Authors

Reviewer 2

Bougea et al present a systematic review of the role of environmental factors in Parkinson's Disease progression.  The authors identified 8 suitable articles and evaluated the following variables' impact on disease progression: exposure to particulate matter of varying sizes, exposure to inhaled compounds, temperature, UV light exposure, and humidity.  While this is certainly a very interesting topic, the very small size of the systematic review (further complicated by poor study quality of the papers included) limits the generalizability and validity of this review.  Please consider the following comments:

  1. The Fig. 1 PRISMA flowchart indicates that 558 records were not retrieved.   Why?  It's ok to exclude reports in accordance to criteria, but those criteria must be justified beyond "not retrieved"... especially if the final study size is only 8 papers.

OUR RESPONSE: We explained in more details that 558 records not  retrieved due to (see figure 1) :

-Methodological design NOT longitudinal prospective

-Irretrievable full document;

- Related only to pharmacological and/or treatment factors (please see the figure)

  1. With such a small number of papers  specific to PD, and given that the hypothesized mechanism of action central to the investigation of environmental factors and PD involves the exacerbation of protein aggregation, it would be prudent to broaden the scope of the investigation to include other disorders that follow a similar pathophysiology to validate its replicability or perhaps to identify a differentiating factor for PD.  

OUR RESPONSE: Actually we are planning to broaden the scope of the investigation to include other disorders such Multiple system atrophy and Dementia with Lewy bodies that follow a similar pathophysiology with PD. However the present review focused on PD was  performed in the context of ALAMEDA project that proposes a novel combination of minimally intrusive wearable devices and smartphone applications required for the desired analysis of motor and non-motor symptoms of PD patients.

Reviewer 3 Report

The study is well-planned and well-described. Still, more is expected from a systematic review - it should not just be a bare report of the included studies’ results. Authors should more comment, compare, assess and/or describe in detail the limitations of the chosen studies… Authors should also suggest some future directions in relation to their conclusions.

Author Response

The study is well-planned and well-described. Still, more is expected from a systematic review - it should not just be a bare report of the included studies’ results. Authors should more comment, compare, assess and/or describe in detail the limitations of the chosen studies… Authors should also suggest some future directions in relation to their conclusions.

OUR RESPONSE: We commented more in detail the limitations of the included studies: ‘’Although the quality assessment of studies included was adequate, there were several limitations. First the cohort size was variable and study design was poor, such as ecological, (only three longitudinal studies). Second, the heterogeneity of data(windows of exposure, lack of adjustment for confounders, absence of sensitive measurement tools, exposure measurement error, follow-up duration) was a major confounding factor. UPDRS was not sensitive to detect seasonal variation of non-motor symptoms such as hallucinations, depression[32,33]. These two studies did not adjust for potential confounding factors that may determine such variation, such as exercise, mood, sleep, and diet. In the same line, another study suggested that an unobserved seasonal variable would need to be highly correlated with temperature, to confound the association between temperature and LED[12]. However, the latter study followed-up 70,000 patients with PD for 23 years[12] ,while the other two studies followed-up comparatively small samples for a short time (2 or 3 years)[33]. Two studies cannot exclude the possibility of potential residual confounding bias[30,31], however, in another study  it is not likely to have occurred[28]. …..In these studies, misclassification of exposure is more likely to result from including patients who do not experience disease aggravation, but are hospitalized due to unrelated health issues[27-29]. By contrast one study used PM2:5 as a sensitive indicator of the disease aggravation[31]……. The evidence of sex differences to air pollution and PD progression/severity  remains unknown, and further investigation and reporting of sex-stratified results will be informative and may shed light into possible biological mechanisms.’’ (lines 12-31 page7)

We also also suggest some future directions in the part of the conclusions :’’ More studies should explore the link and toxicological mechanisms associated with gene-environment interaction and air pollution. Keeping in mind that brain is the most heat-sensitive organ, identifying how temperature may affect the motor and non-motor symptoms of PD would improve our understanding of the pathophysiology of PD. Future studies could examine the interaction between the human gut microbiome and exposure to air pollution with regard to PD risk and possibly progression. Dose-response effects evaluated by high-quality studies are warranted, especially on seasonal effect ultraviolet light on PD progression or severity . …..Now is the time take advantage of the multipollutant statistical methods such as machine learning algorithms, to predict the health effects of air pollution on PD progression. However, by employing a wide variety of multipollutant statistical methods across different epidemiologic study designs, we build up the essential scientific base in order to potentially develop more sustainable, multipollutant air quality regulations. Toward this direction, biomonitoring with plants would be an adequate alternative technique to acquire data about pollution exposure to residential or occupational ambient of patients with PD. ‘’ (lines 9-20 page8)

Reviewer 4 Report

In this review article, the authors summarise what is known about the effects of environmental factors such as air pollution, temperature, humidity, solar radiation and seasonal changes on PD. Based on the inclusion/exclusion criteria, they included 8 studies in their review. The paper addresses an important but under-researched area of PD. I have a few comments and suggestions for improving the paper.

1. They confuse the terms "environmental" and "ecological". I would prefer "enviromental" to "ambiental". However, "ambiental" may be more accurate, as you do not discuss the other, better researched environmental factors that already had a clear link to the development of PD, such as pesticides.

2. In relation to the first question, why did you not consider the impact of pesticides on PD? 

3. I do not like the way the results section is structured. You discuss each individual study, starting for example with Lee et al 2017 - followed by a more or less detailed description of the main results of the study. I would suggest that you leave the detailed information in the tables and simply summarise the data from all the studies in the Results section, followed by a proper discussion of the significance of the results in the next section.

4. I would be a little more careful about the conclusions. For example, in the summary you say: "Specific air pollutants and temperature variations are important factors in the progression of PD ". Neither in your report nor in the experimental studies is there enough evidence to support such a bold claim. Perhaps there is such a link between air pollutants and temperature, but that has yet to be proven. Please, change that.

Author Response

In this review article, the authors summarise what is known about the effects of environmental factors such as air pollution, temperature, humidity, solar radiation and seasonal changes on PD. Based on the inclusion/exclusion criteria, they included 8 studies in their review. The paper addresses an important but under-researched area of PD. I have a few comments and suggestions for improving the paper.

1. They confuse the terms "environmental" and "ecological". I would prefer "enviromental" to "ambiental". However, "ambiental" may be more accurate, as you do not discuss the other, better researched environmental factors that already had a clear link to the development of PD, such as pesticides.
OUR RESPONSE:thank you for the comment. We agree  with the Reviewer to use the ‘’ambiental’’ as it is more accurate as we did not discuss the other, better researched environmental factors that already had a clear link to the development of PD, such as pesticides.

  1. In relation to the first question, why did you not consider the impact of pesticides on PD? 
    OUR RESPONSE: So far there have been published many studies of pesticides on PD risk onset, but not on PD progression. Thus, we focused on the impact of other environmental factors on PD progression on this review.

  2. I do not like the way the results section is structured. You discuss each individual study, starting for example with Lee et al 2017 - followed by a more or less detailed description of the main results of the study. I would suggest that you leave the detailed information in the tables and simply summarise the data from all the studies in the Results section, followed by a proper discussion of the significance of the results in the next section.
    OUR RESPONSE:We summarised the data from all the studies in the Results section, followed by a proper discussion of the significance of the results in the next section, as reviewer suggested.

  3. I would be a little more careful about the conclusions. For example, in the summary you say: "Specific air pollutants and temperature variations are important factors in the progression of PD ". Neither in your report nor in the experimental studies is there enough evidence to support such a bold claim. Perhaps there is such a link between air pollutants and temperature, but that has yet to be proven. Please, changethat.

 OUR RESPONSE: We changed this sentence of the conclusions  as reviewer suggested:’’There is a link between air pollutants and temperature, but that has yet to be proven.’’ (see part of Conclusion in the Abstract)

Round 2

Reviewer 2 Report

Thank you for addressing the original review.  Point 1 has been adequately addressed.  Point 2 should be clarified in the manuscript to quell any potential reader concern for limited scope of investigation.

Author Response

Reviewer 2

Thank you for addressing the original review.  Point 1 has been adequately addressed.  Point 2 should be clarified in the manuscript to quell any potential reader concern for limited scope of investigation.

As concerning the Point2 we clarified the limited scope of investigation in Introduction: ‘’Moreover, the present review focused only in PD and did not  include other disorders such Multiple system atrophy and Dementia with Lewy bodies that follow a similar pathophysiology with PD, because it was performed in the context of ALAMEDA project that proposes a novel combination of minimally intrusive wearable devices and smartphone applications required for the desired analysis of motor and non-motor symptoms of PD patients.’’ (lines 29-34 page2) and in the exclusion criteria: ‘’ those that consider similar disorders with PD such as Multiple system atrophy and Dementia with Lewy bodies’’ (lines 12-13 page 3)

Reviewer 4 Report

No further comments.

Author Response

Comments and Suggestions for Authors: No further comments   OUR RESPONSE: We thank the Reviewer